# Restriction site-associated DNA sequencing (RAD-seq) of tea plant (*Camellia sinensis*) in Sichuan province, China, provides insights into free amino acid and polyphenol contents of tea

Xiaoping Wang[ORCID]*[◎], Minshan Sun[◎], Yuanyuan Xiong, Xiao Liu, Chunhua Li, Yun Wang, Xiaobo Tang

Tea Research Institute, Tea Refining and Innovation Key Laboratory of Sichuan Province, Sichuan Academy of Agricultural Sciences, Chengdu, PR China

◎ These authors contributed equally to this work.

* wangxp922@sina.com

**Data Availability Statement:** The datasets generated and analyzed during the current study

## Abstract

Worldwide, tea is a popular beverage; within the realm of Chinese tea, Sichuan tea holds particular significance for its role in the origin and composition of Chinese tea cultivars. Sichuan tea is noted for its rich content of free amino acids (FAAs) and tea polyphenols (TPs), which has made it an important subject for studying genetic diversity and the genes regulating these compounds. In this study, 139 varieties of tea were collected from areas in Sichuan Province, China, with similar geographical and climatic conditions. The FAA content was approximately 3% and the TP content was approximately 17%. Using RAD sequencing, 5,656,224 variant loci were identified, primarily comprising SNPs (94.17%) and indels (5.83%). Evolutionary analysis revealed that genetic divergence was not closely linked to the collection location. Population structure analysis confirmed a division into two main populations having a similar composition to the phylogenetic clusters. Screening for FAA-related SNPs identified significant loci associated with 33 genes that potentially regulate FAA content. Similarly, TP-related analysis pinpointed 8 SNPs significantly linked to 20 candidate genes. Notably, genetic associations hinted at the genes involved in the stress response and the accumulation of phenolic compounds, enhancing the understanding of determinants of tea quality. This research underscores the potential for molecular breeding based on genetic insights, suggesting pathways to improve the FAA and TP contents in tea. These findings not only provide a solid foundation for exploring gene–chemical interactions but also offer practical strategies for improving the nutritional and sensory attributes of tea cultivars through informed breeding practices.

are available in the NCBI Sequence Read Archive (SRA) repository (accession no. SRR28840134-SRR28840272).

**Funding:** This work was supported by Tea varieties breeding and promotion project of Sichuan (sccxtd-2023-10 to WXP), http://nynct.sc.gov.cn/; Natural science Foundation of Sichuan (2023NSFSC0163 to WXP), https://kjt.sc.gov.cn/; Financial independent project of Sichuan (2022ZZCX055 to WXP), http://www.chinawestagr.com/index.asp;"The 14th Five-Year Plan" tea tree breeding project of Sichuan (2021YFYZ0025 to WY), https://kjt.sc.gov.cn/; "1 + 9" scientific and technological research project of Sichuan Academy of Agricultural Sciences (no grant numbers. to WY), http://www.chinawestagr.com/index.asp. The funders had no role in study design, data collection and analysis, decision to publish, or preparation of the manuscript.

**Competing interests:** The authors have declared that no competing interests exist.

## Introduction

Tea, a magical beverage originating from ancient China, is accompanied by thousands of years of culture and history [1]. At present, tea, as an important economic crop in Asia, Africa, America, and elsewhere, has been widely developed and planted, and has an important contribution to the agricultural economy [2]. In China, tea plants are primarily cultivated in the southwest, south of the Yangtze River, and north of the Yangtze River regions [3, 4]. Following the selection and cultivation of specific flavors and aromas, several renowned tea cultivars have emerged, including "Longjing," "Dahongpao," and "Pu-erh."

Sichuan is a major tea-producing area in China in which Ya'an, Chengdu, and Emeishan are famous varieties of tea [4]. In Sichuan tea, the original color of the tea leaves is usually maintained, providing a crisp taste and unique aroma. The rich resource of tea cultivars in Sichuan provides a basis for the further cultivation of even better varieties, and facilitate the study of the molecular mechanism of the formation of the unique flavor of Sichuan tea.

Tea is rich in nutrients, including free amino acids (FAAs), tea polyphenols (TPs), and vitamins. FAAs primarily contribute to the refreshing taste of tea, not only endowing tea with its unique flavor but also exerting vital physiological functions in the human body [5–8]. For example, theanine has antidepressant and mood-boosting effects, providing a significant role in improving people's mental health [9–12]. TPs are the primary cause of the antioxidant properties of tea, capable of scavenging free radicals, delaying aging, and preventing various chronic diseases [13, 14]. Additionally, TPs exhibit antibacterial and antiviral bioactivities, which hold considerable significance for maintaining human health [15–18].

The advent of high-throughput technology has facilitated the widespread application of restriction site-associated DNA sequencing (RAD-seq) in the development of molecular markers associated with desirable traits in a diverse array of plants [19–25]. Currently, the genomic data for tea plant have been completed, and by investigating the data of target traits, combined with genome-wide association analysis (GWAS), the molecular marker sites and candidate genes related to traits can be rapidly identified [2, 26–30]. Qiu *et al.* used GWAS to investigate the diversity of flavonoid metabolites in tea [31]. Through screening, Lu *et al.* identified four genes related to the plant size and leaf color of tea plant using GWAS. The results indicated that these genes were associated with the size, leaf color, *etc* [32]. Jiang *et al.* employed GWAS to identify aroma substances in 70 tea cultivars from Fujian province and Liu *et al.* revealed the phylogenetic relationships of four typical tea landraces from Hunan province [33–35].

In this study, we collected 139 commonly used tea cultivars in Sichuan. We measured the concentrations of FAAs and TPs within these samples. Subsequently, we employed restriction site associated DNA sequencing (RAD-seq) and GWAS to explore the genetic evolution of these teas, as well as the identification of molecular markers and candidate genes related to the FAA and TP content in Sichuan teas. The results provide a reference for the study of flavor differences in Sichuan tea cultivars, as well as a scientific reference for the more accurate localization of key genes for tea flavor formation and variety breeding.

## Materials and methods

### Plant materials

From plants cultivated under the same soil and environmental conditions in Meishan City, Sichuan province (29.81˚N, 103.17˚E), 139 tea plant accessions were collected for RAD-seq construction. The geographic origins of these accessions were mainly from different areas of the Sichuan province, China, and comprised 91 from Yaan, 9 from Chengdu, 8 from Meishan, 14 from Guangyuan, and 17 from Mianyang (S1 Table). In the spring of April, buds with two

tea plant leaves showing good growth were taken as materials, and the samples were stored in the refrigerator at −80°C after being frozen in liquid nitrogen.

## Detection of FAAs and TPs

The total FAA content in tea was determined using ninhydrin colorimetry (GB/T 8314–2013) [36]. The detection of TPs in tea was performed using the Folin phenol method referenced in the National Standards of China (GB/T 8313–2018) [37]. Normal distribution analysis with Kolmogorov-Smirnov test was performed, histogram and violin plot were drawn by GraphPad Prism 9.5.0.

## DNA extraction and sequencing

Genomic DNA samples from all accessions were extracted using a CTAB method and incubated for 1 hour at 37°C with 2 μL of RNase-A [38]. The genomic DNA was digested with the restriction enzyme *Eco*RI with the addition of the P1 adapter. The ligated products were then pooled, broken into small fragments, and fragments of 300–700 bp were separated using 2% agarose gel electrophoresis. The P2 adapter was added and PCR amplification was performed to enrich RAD tags. The DNA libraries were using the Illumina sequencing platform.

## Data processing

The sequencing original data from the Illumina platform were screened using fastp (version 0.18.0) [39]. Reads with unknown nucleotides (N) $\geq$ 10%, Phred quality score $\leq$ 20 with more than 50% base content, and those containing connectors were excluded. Subsequently, the filtered reads were compared with the reference genome using BWA software (version 0.7.12) [40]. The results were sorted using Sambamba and repeated sequences were labeled [41]. The reference *C. sinensis* genome was downloaded from the BIG database (accession no.: GWHACFB00000000) [2]. Variants were detected by GATK (version 4.0) with the filter parameters (−Window 4, −filter "QD < 4.0 || FS > 60.0 || MQ < 40.0 ", −G_filter "GQ < 20") [42, 43].

## Population structure, principal component analysis, and phylogenetic tree construction

The population structure of 139 tea plants was conducted using Admixture (version 1.3.0), and the tested K was set from 1 to 9 [44]. According to the cross-validation method, the best genetic cluster, K, was determined and the results were plotted using R. Pophelper software (v2.2.7) was used to plot the genetic composition of each sample [45]. PCA and kinship was analyzed using GCTA (version 1.92.2) [46]. A phylogenetic tree was conducted using MEGA (version 7.0) and the NJ algorithm with 500 bootstrap replicates.

## Genome-wide association analysis

Results with a heterozygosity ratio greater than 0.8 and a minor allele frequency (MAF) less than 0.05 were removed. The decay of linkage disequilibrium (LD) was evaluated using PopldDecay software [47]. A genome-wide association study (GWAS) was performed based on the GLM, GLM(Q), MLM(K), and MLM(QK) models to calculate the relationships between markers and phenotyping data using the GEMMA (v0.98.1) [48].

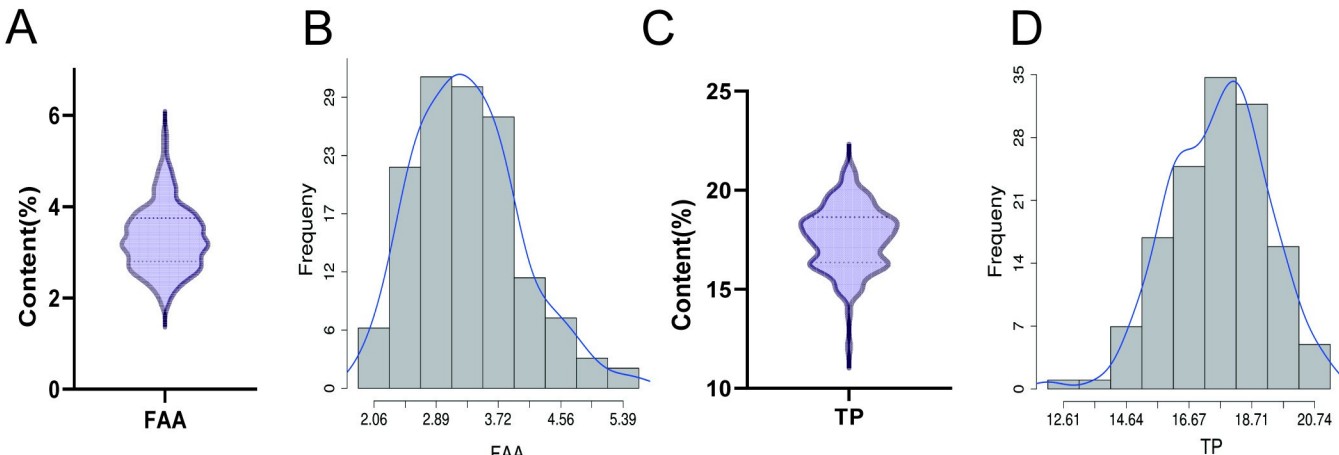

**Fig 1. The frequency distribution histogram of FAAs and TPs.** (A) FAA content distribution. (B) Frequency distribution histogram of FAA. (C) TP content distribution. (D) Frequency distribution histogram of TP.

## Results

### Phenotypic analysis of free amino acids and tea polyphenols

The 139 varieties were all collected from the Sichuan Province of China, mainly within 8 areas with relatively similar geographic and climatic environments. The FAA and TP contents of these 139 varieties all showed a normal distribution, with the FAA content mainly distributed at approximately 3% and the TP content mainly distributed at approximately 17% (Fig 1A–1D).

### Statistics of sequencing data

The *Eco*RI cleavage sites contained in the 15 chromosomes of the tea plant genome were evaluated, with a total of 1,274,788 sites found, of which the number of effective cleavage sites was 866,855 (Fig 2A, S2 Table). After sequencing 139 samples, we obtained 511 Gb of clean reads, with an average of more than 3 Gb of data per sample, and the Q30 values of the samples were all approximately 90% (S3 Table). The mapping rate of the obtained clean reads was greater than 96% for all sample data after matching to the reference genome (S4 Table).

Through the analysis, we screened a total of 5,656,224 variant loci, including 5,326,438 SNP loci (94.17%) and 329,786 indel loci (5.83%). The two most prominent functions of loci were nonsynonymous SNVs (88,990) and synonymous SNVs (80,574) (Fig 2B). These SNPs were mainly distributed in the intergenic region, followed by the intronic region, and only 175,323 were distributed in the exonic region, in addition to a small number of SNPs distributed in the upstream and downstream regions of the gene (Fig 2C). Among these SNPs, there were 3,779,918 SNPs in the transition category, mainly in the G->A and C->T categories, and there were 1,546,520 SNPs in the transversion category (Fig 2D).

### Evolutionary analysis

After filtering for rare alleles, high deletion rates, and high heterozygosity, we retained 1,179,501 variation loci, including 1,107,524 SNP loci and 71,977 indel loci. The phylogenetic analysis of these individuals showed that these varieties diverged depending on the collection location, suggesting that they may have a relatively similar have genetic background (Fig 3A). The analysis of their components showed that all varieties were mainly divided into two groups

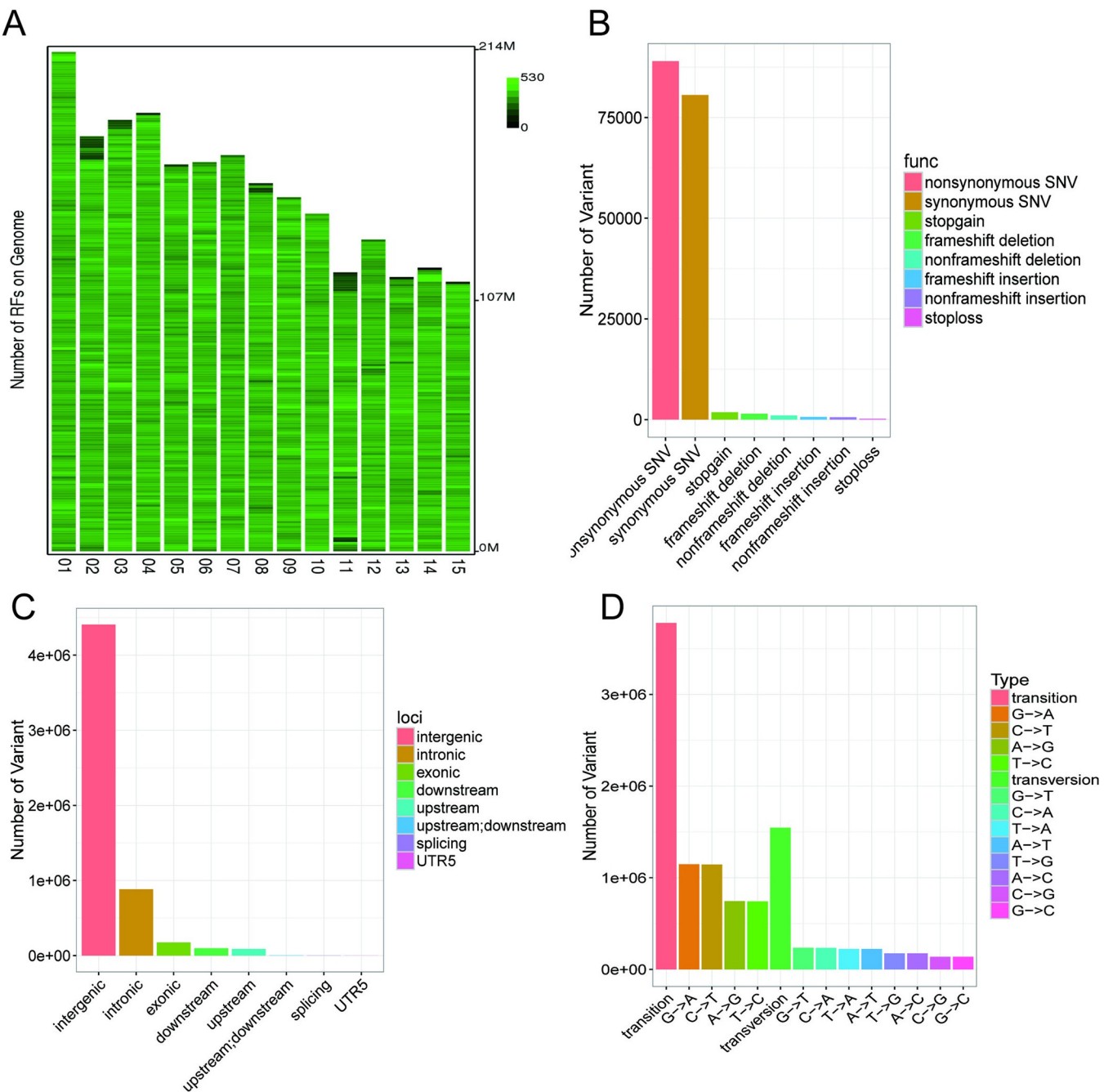

**Fig 2. The distribution of RF and the analysis of SNPs and indels.** (A) Chromosome distribution map of RF. (B) The location information of SNPs and indels. (C) The functional annotation of SNPs and indels. (D) Transition and transversion statistics of SNPs.

(Fig 3B), with group I containing approximately 100 varieties and group II containing approximately 33 varieties, and a further 5 varieties clustered into cluster III (Fig 3A and 3B). The results of the linkage disequilibrium (LD) decay values of all the populations showed that the $R^2$ value decreased rapidly from 0.1 to 0.05 in the 0–50 kb range, indicating the high precision of the correlation analysis results (Fig 3C).

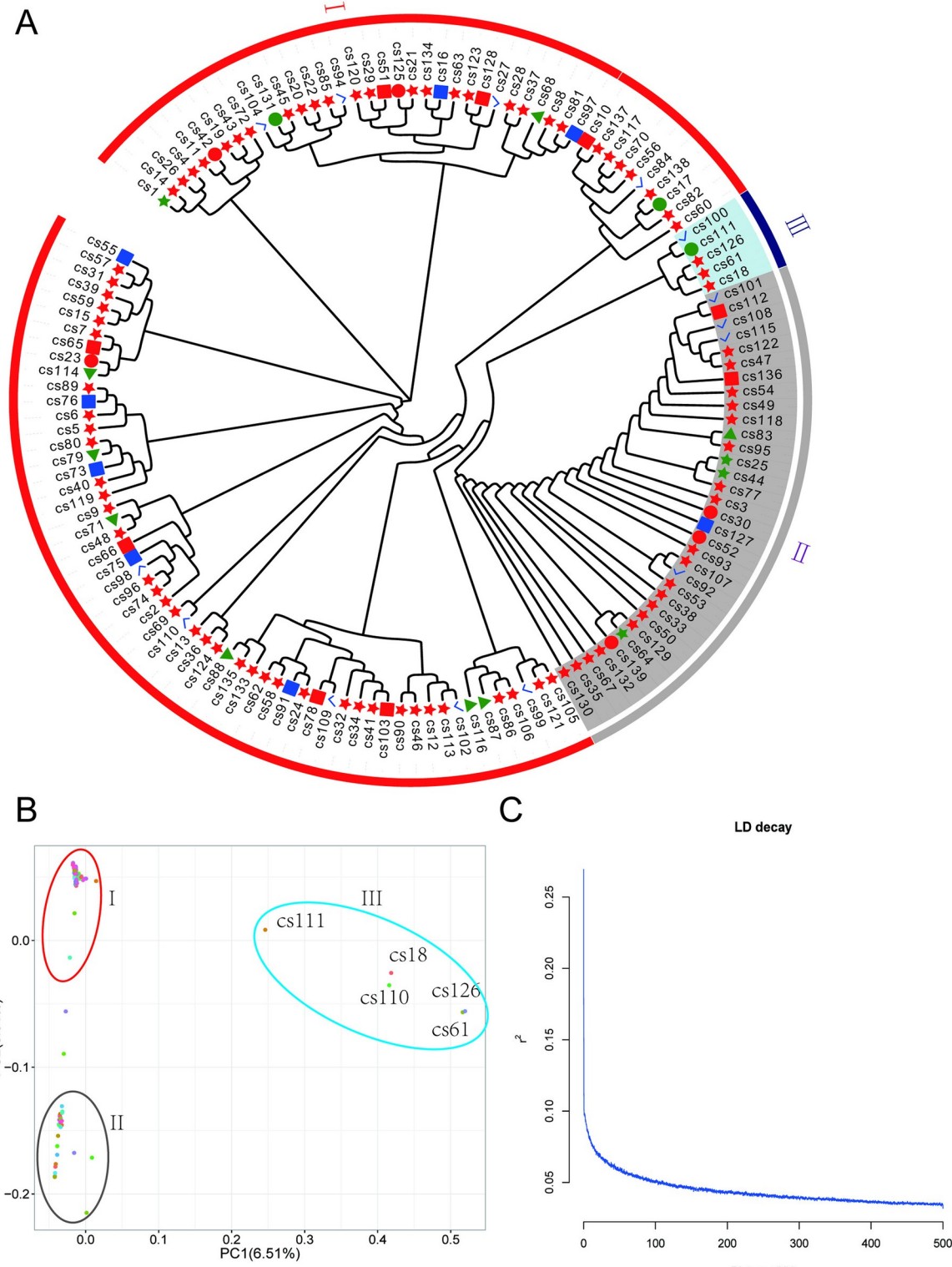

**Fig 3. The phylogenetic evolution tree and principal component of the all accessions.** (A) Evolutionary tree. Green circle, CDCZ. Red circle, CDQL. Blue check, GYQC. Green triangle, MSHY. Red rectangle, MYAX. Red star, YAMS. Green star, YAYJ. (B) Principal component analysis (PCA) of 139 tea plant cultivars. (C) LD decay analysis. GYQC, Guangyuan Qingchuan. MYBC, Mianyang Beichuan. MYAX, Mianyang Anxian. CDQZ, Chengdu Chongzhou. CDQL, Chengdu Qionglai. YAMS, Yaan Mingshan. YAYJ, Yaan Yajiang. MSHY, Meishan Hongya.

## Population structure analysis

The values of the cross-validation (CV) errors were calculated for each value of K from 2 to 9. The results showed that all accessions could be divided into two populations (K = 2), showing similar results to the PCA and evolutionary analysis (Fig 4A and 4B). When K = 2, the composition of the two populations corresponded essentially to clusters I and II in the phylogenetic tree, and when K = 2, the compositional varieties of the three populations corresponded basically to the three clusters in the evolutionary tree (Figs 3A and 4C).

## Screening of FAA-related SNPs and candidate genes

We analyzed the SNPs associated with the FAA contents of four models, and the results showed that under p < 0.00001 conditions, the four methods identified a total of 30 significant SNP loci, of which 14 SNP loci were shared (Fig 5A). The genes within 50 kb of these significantly related SNP loci were analyzed, and the results showed that there were 33 related genes (Fig 5B). These loci may be significantly associated with the FAA content, and these candidate genes were distributed on six chromosomes and two scaffolds: Chr1 with one SNP locus and the associated genes, including Cha01g016240 (prolyl 4-hydroxylase 10, P4H10), Cha01g016250, Cha01g016260, Cha01g016270 (phospholipase D delta, PLDDELTA), and Cha01g016280 (serine/arginine-rich splicing factor SR45a, SR45a); Chr3 with one SNP locus associated with three candidate genes (Cha03g018190, Cha03g018200, Cha03g018210); Chr8 with one SNP locus associated with Cha08g000060, Cha08g000070, Cha08g000080 (5-methyltetrahydropteroyltriglutamate–homocysteine methyltransferase, METE); Chr9 with one SNP locus associated with Cha09g016500, Cha09g016510, Cha09g016520 (nuclear transcription factor Y subunit B-5, NFYB5), Cha09g016530 (cullin-1, CUL1); three significant SNP sites associated with ten genes on Chr5, such as Cha05g013620 (Lectin receptor-like kinase 4, LecRK4), Cha05g013630 (LecRK4), Cha05g013660 (LecRK3) (Fig 5C and 5D, and S5 Table).

## Screening of TP-related SNPs and candidate genes

The same analysis of the SNPs associated with TP showed that eight SNPs were significantly associated (p < 0.00001) by four of the methods (Fig 6A). There were 20 genes potentially associated with these SNPs (Fig 6B). The results of the GLM(Q) model showed that these SNPs were mainly distributed onto chromosomes Chr1, Chr2, Chr4, Chr8, Chr9, and Chr11, which were significantly correlated with TPs (Fig 6C and 6D). The genes associated with the SNP loci on chromosome Chr1 were Cha01g002220 (NAC078), Cha01g002230 (NAC078), Cha01g002240 (homology of malate dehydrogenase 1, MDH1), Cha01g002250 (homology of ACYL-CoA-BINDING PROTEIN4, ACBP4), Cha01g002260, and Cha01g006610 (non-specific lipid transfer protein 1) and the genes linked to the SNP loci on chromosomes Chr2 and Chr4 were Cha02g000510 (MYB3) and Cha04g018610 (fluG), respectively; the genes associated with the significant SNP loci on chromosome 8 were Cha08g010570, Cha08g010580, Cha08g010590, and Cha08g010600, and the genes associated with the SNP loci on chromosome Chr9 were Cha09g017840 (histone acetyltransferase, HAT), Cha09g017850 (VESICLE-ASSOCIATED MEMBRANE PROTEIN721, VAMP721), and Cha09g017860 (vacuolar protein sorting 13, VPS13), in addition to the genes associated with the SNP loci on Chr11, Cha11g000470 (glycerol-3-phosphate acyltransferase 3, GPAT3), Cha11g000480 (GPAT3), Cha11g000490, Cha11g000500, Cha11g000510, Cha11g000520, Cha11g000530 (long chain acyl-CoA synthetase 1, LACS1), and Cha11g000540 (LACS1) (S6 Table).

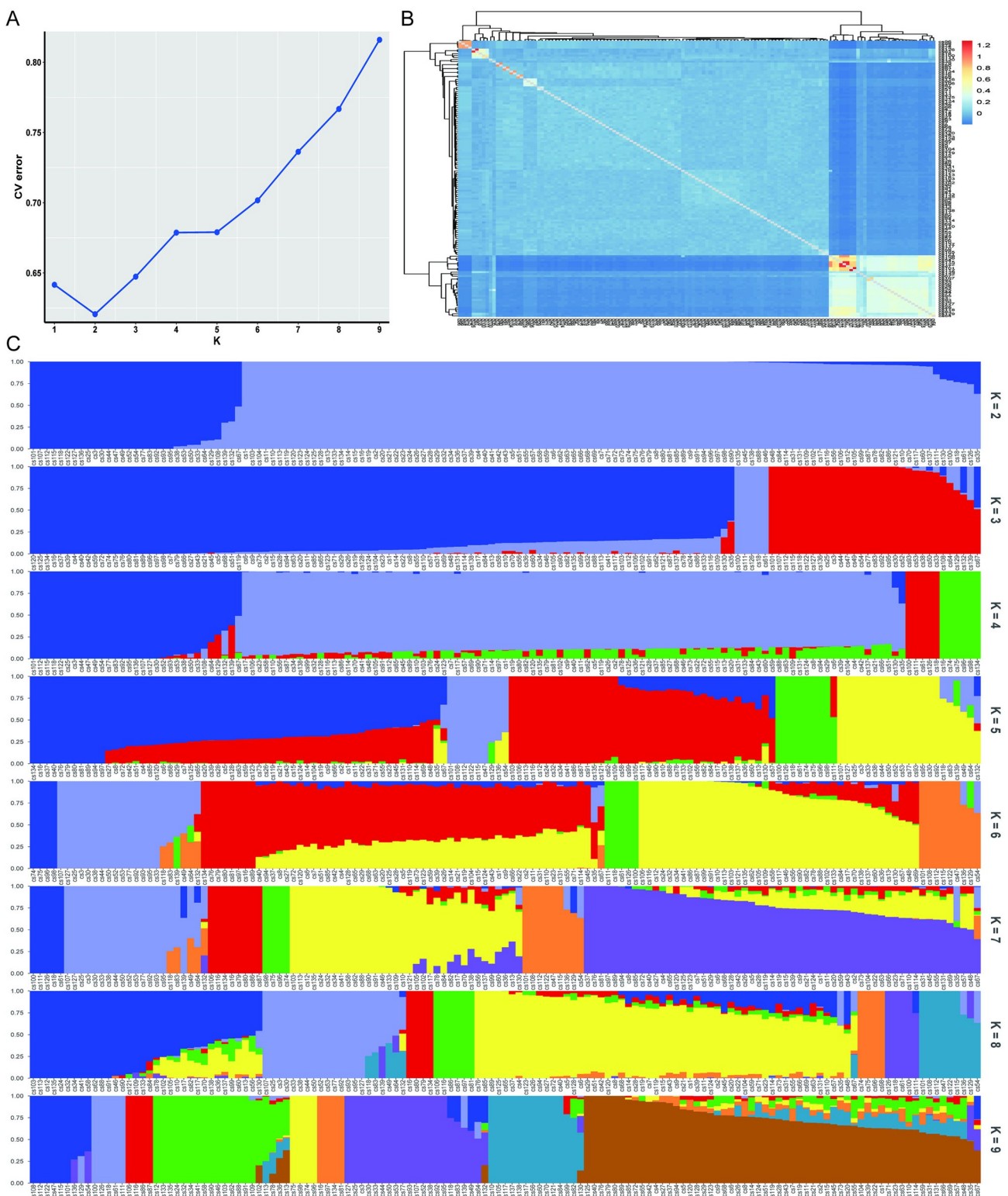

**Fig 4. Population structure of 139 tea plant cultivars.** (A) Cross-validation of K values. (B) Heat map of relationship between samples. (C) Stacked diagram of the tea plant community structure.

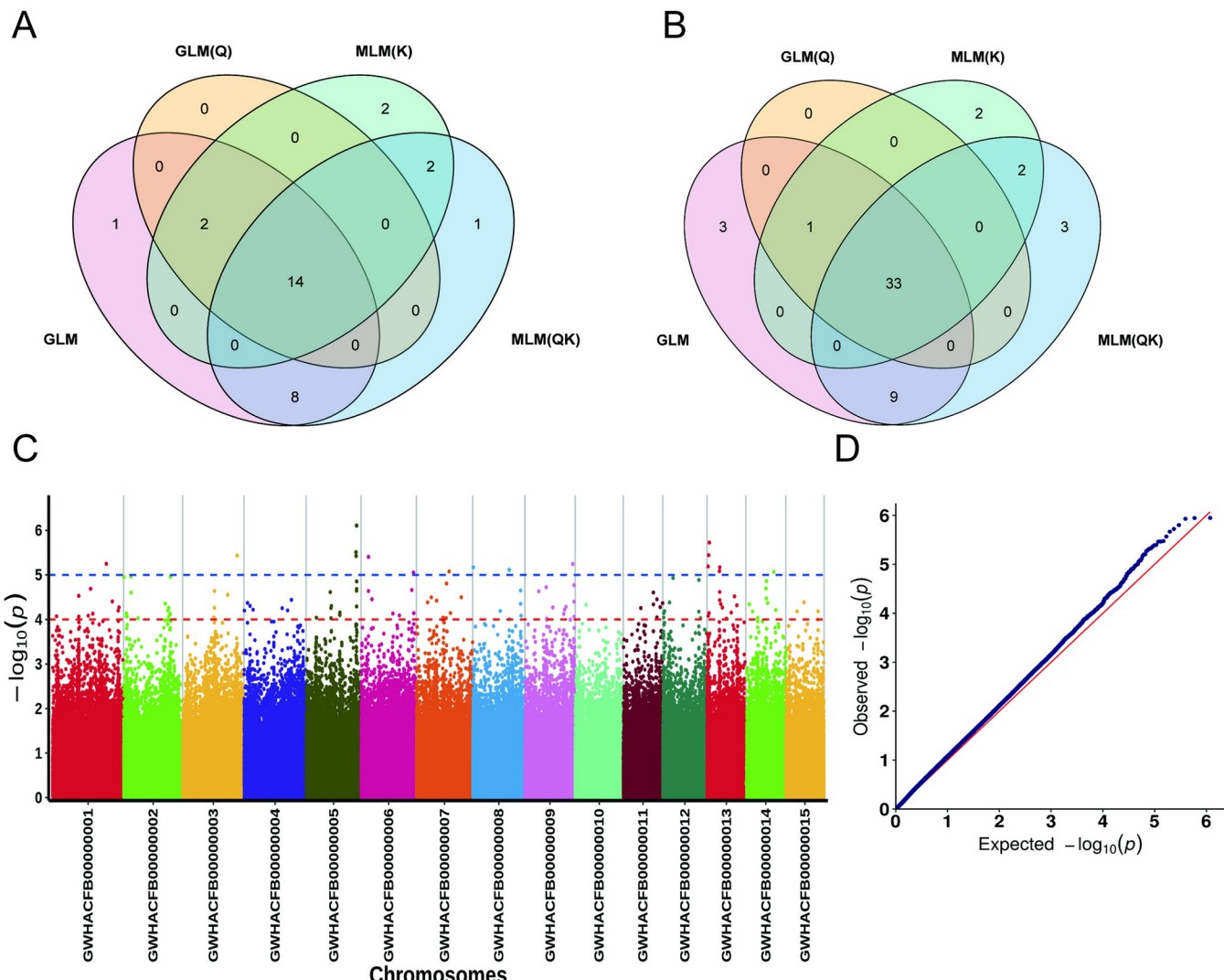

**Fig 5. FAA-related SNPs and candidate genes.** (A) FAA-related SNPs screened from the four models. (B) FAA-related genes screened from the four models. (C) Manhattan plots of FAA association analysis by GLM(Q). Negative log10-transformed *P*-values from a genome-wide scan are plotted against position on each of the 15 chromosomes. Red and black dots indicates the genome-wide significance threshold ($p >= 4$ and 5, respectively). (D) Quantile–Quantile (Q–Q) plots of FAA traits. The red line represents the predicted values and the blue dots represent the observed values, showing the difference between the predicted value and the observed value.

## Discussion

Tea is one of the most important economic crops in China, and Sichuan is a major tea-producing region in southwest China that is the origin of a series of well-known tea cultivars with unique flavors and textures. In this study, RAD sequencing was performed on 139 tea species from several major tea-producing areas in the Sichuan province, China. The analysis of these data can provide useful resources for the collection and molecular breeding of teas from the Sichuan region. The results of these data suggest that the 139 tea accessions from the Sichuan region collected in this study consist of two populations.

The main cultivated tea cultivars belong to the genus *Camellia L.*, section Thea (L.) Dyer, which contains the two major varieties of *C. sinensis* var. *sinensis* and *C. sinensis* var. *assamica* (Masters) Chang [2]. Wang et al. collected multiple accessions in the Sichuan region in their

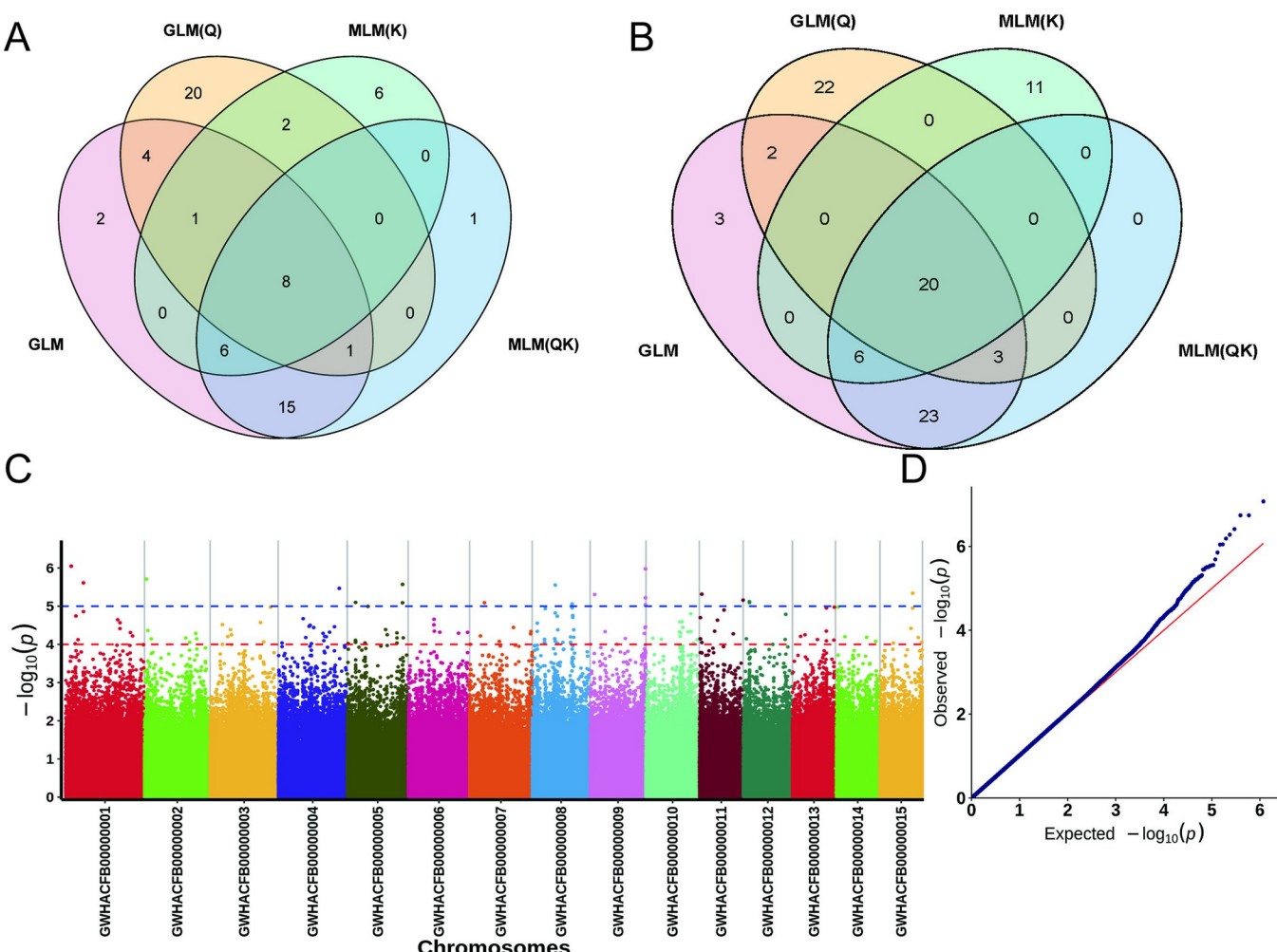

**Fig 6. TP-related SNPs and candidate genes.** (A) TP-related SNPs were screened using four models. (B) FAA-related genes were screened using four models. (C) Manhattan plots of TP association analysis by GLM(Q). Negative log10-transformed *P*-values from a genome-wide scan are plotted against position on each of 15 chromosomes. Red and black dots indicates the genome-wide significance threshold ($p > = 4$ and 5, respectively). (D) Quantile–Quantile (Q–Q) plots of TP trait. The red line represents the predicted values and the blue dots represents the observed values, showing the difference between the predicted value and the observed value.

study, and the results showed that these accessions mainly belonged to *C. sinensis* var. *sinensis*, which may predict that multiple accessions in this study also belong to *C. sinensis* var. *sinensis*. Lu et al. analyzed 120 ancient tea plants collected in Guizhou and Yunnan Provinces and found that the natural environment or artificial breeding selection had less selection pressure on ancient tea plants [32]. In this study, we were unable to collect samples from other tea-producing regions for further analysis. The collection sites of multiple accessions in this study were mainly located in hilly areas with altitudes usually of 800–1,500 m. Most of the plants and growing conditions were similar to those of *C. sinensis* var. *sinensis*, which usually has smaller leaves, cold tolerance, and has similar characteristics when growing as a shrub [27, 49, 50]. Jiang et al. analyzed 70 tea cultivars, mainly from Fujian Province, and shows that tea in Fujian is mainly composed of three groups [33]. In terms of population structure and evolutionary relationships, the tea cultivars in Sichuan province mainly diverged into two populations with a highly similar genetic background and typical regional characteristics (Figs 3 and 4).

The contents of total catechins, caffeine, FAA, TP, vitamins, and other components in tea have a direct influence on the flavor of tea, and systematic analyses of FAA and TP contents of tea in Sichuan will provide guidance for breeding for the high content of these two substances. A systematic study of the contents of FAA and TP in tea will guide the breeding of tea with high contents of these two substances. GWAS is a commonly used analytical method, and, in this study, we found that the results obtained from four different analytical models had a high degree of similarity, which indicated a high degree of confidence in the results.

We selected 33 genes based on the FAA analysis results, but the functions of multiple genes are currently unknown. Among them, we found three LECRK homologous genes on chromosome Chr 5. In multiple plant species, LECRK is associated with salt stress and cold stress [51–54]. Furthermore, on chromosome 13, the PGIP1 gene is closely associated with biotic and abiotic stress in multiple species [55–57]. The varieties used in this study were collected from higher altitude regions and belong to *C. sinensis* var. sinensis. They have a strong ability to tolerate cold stress, which suggests a relationship between the FAA content in tea leaves and cold stress.

The GWAS results of TPs revealed 24 candidate genes, including genes homologous to MYB3 on chromosome Chr2, which has been reported to activate the phenylpropanoid pathway and promote the accumulation of polyphenolic substances in tobacco [58]. LACS is a key enzyme involved in the activation of fatty acids into the corresponding CoA esters [59]. Fatty acyl-CoA intermediates may alter the biosynthetic pathways of polyphenolic compounds [60, 61].

## Conclusions

In this study, we collected 139 tea accessions from Sichuan Province, China, and studied the distribution of their SNPs using RAD-seq. The results revealed that these 139 accessions were mainly divided into two genetic populations. The GWAS analysis of FAA and TP contents and SNPs identified 14 SNPs significantly associated with FAAs, and identified 33 candidate genes. Moreover, 8 SNPs were significantly associated with TPs, along with 20 candidate genes. These findings provide a reference for further research into the regulatory relationship between these genes and the content of FAAs and TPs in tea, offering a genetic resource for future molecular breeding.

## Supporting information

**S1 Table. The list of 139 accessions information.**
(XLSX)

**S2 Table. Evaluation of genomic enzyme digestion.**
(XLSX)

**S3 Table. Sequence data statistical.**
(XLS)

**S4 Table. Mapping data statistical.**
(XLS)

**S5 Table. The genes related to FAA.**
(XLSX)

**S6 Table. The genes related to TP.**
(XLSX)

## Acknowledgments

We are grateful to the Henan Assist Research Biotechnology Co., Ltd (Zhengzhou, China) for assisting in sequencing and bioinformatics analysis.

## Author Contributions

**Conceptualization:** Xiaoping Wang.

**Data curation:** Minshan Sun, Xiao Liu.

**Investigation:** Xiaoping Wang.

**Methodology:** Xiaoping Wang.

**Writing – original draft:** Xiaoping Wang, Yuanyuan Xiong, Chunhua Li, Yun Wang, Xiaobo Tang.

**Writing – review & editing:** Xiaoping Wang, Minshan Sun.

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
