## [Decision Letter · Decision Letter 0]

3 Jul 2024

PONE-D-24-22074Restriction site-associated DNA sequencing (RAD-seq) of tea plant (Camellia sinensis) in Sichuan province, China, provides insights into free amino acid and polyphenol contents of teaPLOS ONE

Dear Dr. Wang,

Thank you for submitting your manuscript to PLOS ONE. After careful consideration, we feel that it has merit but does not fully meet PLOS ONE’s publication criteria as it currently stands. Therefore, we invite you to submit a revised version of the manuscript that addresses the points raised during the review process.

We look forward to receiving your revised manuscript.

Kind regards,

Rajappa Janyanaik Joga, PhD

Academic Editor

PLOS ONE

Journal Requirements:

This work was supported by Tea varieties breeding and promotion project of Sichuan (sccxtd-2023-10), Natural science Foundation of Sichuan (2023NSFSC0163), Financial independent project of Sichuan (2022ZZCX055), “The 14th Five-Year Plan” tea tree breeding project of Sichuan (2021YFYZ0025), “1 + 9” scientific and technological research project of Sichuan Academy of Agricultural Sciences.

4. Thank you for uploading your study's underlying data set. Unfortunately, the repository you have noted in your Data Availability statement does not qualify as an acceptable data repository according to PLOS's standards.

6. We note that Figure 1A in your submission contain map images which may be copyrighted. All PLOS content is published under the Creative Commons Attribution License (CC BY 4.0), which means that the manuscript, images, and Supporting Information files will be freely available online, and any third party is permitted to access, download, copy, distribute, and use these materials in any way, even commercially, with proper attribution. For these reasons, we cannot publish previously copyrighted maps or satellite images created using proprietary data, such as Google software (Google Maps, Street View, and Earth). For more information, see our copyright guidelines: http://journals.plos.org/plosone/s/licenses-and-copyright.

We require you to either present written permission from the copyright holder to publish these figures specifically under the CC BY 4.0 license, or remove the figures from your submission:

a. You may seek permission from the original copyright holder of Figure 1A to publish the content specifically under the CC BY 4.0 license.  

Reviewers' comments:

Reviewer's Responses to Questions

**Comments to the Author**

1. Is the manuscript technically sound, and do the data support the conclusions?

Reviewer #1: Partly

Reviewer #2: Yes

Reviewer #3: Yes

2. Has the statistical analysis been performed appropriately and rigorously? 

Reviewer #1: Yes

Reviewer #2: No

Reviewer #3: I Don't Know

3. Have the authors made all data underlying the findings in their manuscript fully available?

Reviewer #1: Yes

Reviewer #2: No

Reviewer #3: Yes

4. Is the manuscript presented in an intelligible fashion and written in standard English?

Reviewer #1: Yes

Reviewer #2: Yes

Reviewer #3: Yes

5. Review Comments to the Author

**Reviewer #1:** The MS reported RAD-seq of tea plant in Sichuan province, China, providing insights into free amino acid and polyphenol contents of tea. It is acceptable for publication in PLoS One after some revisions.

1.Authoritative permission is requested to use the map in the publication.

2.The FAA content was approximately 3 µg/mL and the TP content was approximately 17 µg/mL. Please check the unit carefully, they should be 30 and 170 µg/mL, respectively. TP is a mixture of phenolic substance, it is better to analyze the main catechins. The GWAS results will be more specific and useful.

3.All tea tree to be tea plant. And tea cultivar is better than tea variety. The 139 accession is recommended to list as Supporting information.

**Reviewer #2:** This study on the genetic basis of the cumulation of FAAs and TP was very significant. The experiment design was reasonable and the research method was advanced too, so the conclusion was reliable.

However the following questions needs further solving.

1. In the section materials and methods, the data analysis methods of the contents of FAAs and TP needs adding. The statistical methods include ANOVA and multiple comparisons, so the section results of the contents of FAAs and Tp needs modifing accordingly.

2. It had better to transfer the figure 1A to the section materials and methods.

3. Add the important pathways of the FAAs metabolism and TP metabolism.

**Reviewer #3:** In general term, the manuscript titled “Restriction site-associated DNA sequencing (RAD-seq) of tea plant (Camellia sinensis) in Sichuan province, China, provides insights into free amino acid and polyphenol contents of tea” represent interesting topic and might be of interest for the broader readers of Plos One.

The authors collected 139 tea varieties from Sichuan, China and analyzed their free amino acids (FAAs) and tea polyphenols (TPs) contents and identified their variant loci using RAD. The results hold interesting data for molecular breeders in tracing their practices to the accumulated levels pf FAAs and TPs. However, the demonstration of the results is below average, and the discussion is generally lacking and superficial. Further, the quality of figures is too low, and the content is rarely readable. I would recommend accept their work after addressing these issues.

6. PLOS authors have the option to publish the peer review history of their article (what does this mean?). If published, this will include your full peer review and any attached files.

Reviewer #1: No

Reviewer #2: **Yes: **Kaibing Zhou

Reviewer #3: No

---

## [Author Response · Author response to Decision Letter 0]

27 Jul 2024

Dear Editors and Reviewers:

Thank you for your feedback and the insightful comments from the reviewers regarding our manuscript. These comments have proven to be valuable in revising and enhancing our MS, as well as providing significant guidance for our research. We have thoroughly analyzed the comments and made appropriate revisions. Now, we resubmit our revised MS, hoping that they will be satisfactory. If you have any queries, please don’t hesitate to contact me at the address below. Moreover, we would like to present the key revisions made in the paper along with our responses to the reviewers' comments listed below, and all revisions showed in tracked MS.

Thank you and best regards.

Yours sincerely,

Xiaoping Wang

Journal Requirements:

A: we have changed MS to PLOS ONE's style requirements.

A: This study is not engaged in the study of the genetic diversity of the tea tree, there are no ethical issues involved and no authorization is required.

This work was supported by Tea varieties breeding and promotion project of Sichuan (sccxtd-2023-10), Natural science Foundation of Sichuan (2023NSFSC0163), Financial independent project of Sichuan (2022ZZCX055), “The 14th Five-Year Plan” tea tree breeding project of Sichuan (2021YFYZ0025), “1 + 9” scientific and technological research project of Sichuan Academy of Agricultural Sciences.

A: Financial Disclosure Statement are revised and submitted as an additional file.

4. Thank you for uploading your study's underlying data set. Unfortunately, the repository you have noted in your Data Availability statement does not qualify as an acceptable data repository according to PLOS's standards.

A: we have changed the accession to “The datasets generated and analyzed during the current study are available in the NCBI Sequence Read Archive (SRA) repository (accession no. SRR28840134-SRR28840272).

”

A: we have added ORCID iD of corresponding author (0009-0003-2006-8467).

6. We note that Figure 1A in your submission contain map images which may be copyrighted. All PLOS content is published under the Creative Commons Attribution License (CC BY 4.0), which means that the manuscript, images, and Supporting Information files will be freely available online, and any third party is permitted to access, download, copy, distribute, and use these materials in any way, even commercially, with proper attribution. For these reasons, we cannot publish previously copyrighted maps or satellite images created using proprietary data, such as Google software (Google Maps, Street View, and Earth). For more information, see our copyright guidelines: http://journals.plos.org/plosone/s/licenses-and-copyright.

A: Thanks for your comments. We removed Figure 1A due to the inability to obtain authorization for the map, which has no impact on the reader's understanding of the main content of the paper.

Reviewers' comments:

Reviewer's Responses to Questions

Comments to the Author

Reviewer #1: The MS reported RAD-seq of tea plant in Sichuan province, China, providing insights into free amino acid and polyphenol contents of tea. It is acceptable for publication in PLoS One after some revisions.

1. Authoritative permission is requested to use the map in the publication.

A: Thanks for your comments. We removed Figure 1A due to the inability to obtain authorization for the map, which has no impact on the reader's understanding of the main content of the paper.

2.The FAA content was approximately 3 µg/mL and the TP content was approximately 17 µg/mL. Please check the unit carefully, they should be 30 and 170 µg/mL, respectively. TP is a mixture of phenolic substance, it is better to analyze the main catechins. The GWAS results will be more specific and useful.

A: Thanks for your comments. After careful checking, it was found that the wrong unit was used and the correct unit should be the ratio of the score to the mass of the dry matter.

We have changed “µg/mL” to “%”.

3.All tea tree to be tea plant. And tea cultivar is better than tea variety. The 139 accession is recommended to list as Supporting information.

A: Thanks. we changed ‘tea tree’ to ‘tea plant’, and ‘tea varieties’ to ‘tea cultivars’. The 139 accessions information are list in S1Table. And change the original S1-S5 Table to S2-S6 Table in turn.

Reviewer #2: This study on the genetic basis of the cumulation of FAAs and TP was very significant. The experiment design was reasonable and the research method was advanced too, so the conclusion was reliable.

However the following questions needs further solving.

1. In the section materials and methods, the data analysis methods of the contents of FAAs and TP needs adding. The statistical methods include ANOVA and multiple comparisons, so the section results of the contents of FAAs and Tp needs modifing accordingly.

A: Thanks. in this study, we mainly the normal distribution of FAAs and TP contents. We added the methods “Normal distribution analysis was performed; histogram and violin plot were drawn by GraphPad Prism 9.5.0.

2. It had better to transfer the figure 1A to the section materials and methods.

A: Thanks. We removed Figure 1A due to the inability to obtain authorization for the map, which has no impact on the reader's understanding of the main content of the paper.

3. Add the important pathways of the FAAs metabolism and TP metabolism.

A: Thanks for your suggestion. Given that "FAAs" and "TPs" are used as general terms to refer to a large group of metabolites, respectively. The metabolic pathways of each metabolite may differ, and this information is not currently available.

Reviewer #3: In general term, the manuscript titled “Restriction site-associated DNA sequencing (RAD-seq) of tea plant (Camellia sinensis) in Sichuan province, China, provides insights into free amino acid and polyphenol contents of tea” represent interesting topic and might be of interest for the broader readers of Plos One.

The authors collected 139 tea varieties from Sichuan, China and analyzed their free amino acids (FAAs) and tea polyphenols (TPs) contents and identified their variant loci using RAD. The results hold interesting data for molecular breeders in tracing their practices to the accumulated levels pf FAAs and TPs. However, the demonstration of the results is below average, and the discussion is generally lacking and superficial. Further, the quality of figures is too low, and the content is rarely readable. I would recommend accept their work after addressing these issues.

A: Thanks for your suggestion. In this study, we collected 139 accessions and analyzed the SNPs associated with FAAs and TPs as well as the genes that might be associated. The results provide some reference. Nonetheless, the results are still relatively scarce and unconvincing due to the lack of further research and proof of the functions of the candidate genes. In addition, although we obtained the cultivar name and geographic information of the collected tea varieties, the genetic information of the formation of these varieties is still lacking, which makes it difficult to analyze the origin and kinship of these varieties in depth. All figures in this thesis are provided as tif files at 300 ppi resolution, and we can also provide them in PDF format if higher quality images are required. We have further revised the paper based on the comments of reviewers #1, #2 and your comments, and hope that this version is more readable.

Others:

P2, we changed “Jiang et al. employed GWAS to identify aroma substances in 70 tea cultivars from Fujian” to “Jiang et al. employed GWAS to identify aroma substances in 70 tea cultivars from Fujian province and Liu et al. revealed the phylogenetic relationships of four typical tea landraces from Hunan province” 

We added five important literature citations.

---

## [Decision Letter · Decision Letter 1]

6 Nov 2024

Restriction site-associated DNA sequencing (RAD-seq) of tea plant (Camellia sinensis) in Sichuan province, China, provides insights into free amino acid and polyphenol contents of tea

PONE-D-24-22074R1

Dear Dr. Wang,

We’re pleased to inform you that your manuscript has been judged scientifically suitable for publication and will be formally accepted for publication once it meets all outstanding technical requirements.

Kind regards,

Mojtaba Kordrostami, Ph.D.

Academic Editor

PLOS ONE

Additional Editor Comments (optional):

Reviewers' comments:

Reviewer's Responses to Questions

**Comments to the Author**

1. If the authors have adequately addressed your comments raised in a previous round of review and you feel that this manuscript is now acceptable for publication, you may indicate that here to bypass the “Comments to the Author” section, enter your conflict of interest statement in the “Confidential to Editor” section, and submit your "Accept" recommendation.

Reviewer #1: All comments have been addressed

Reviewer #2: All comments have been addressed

2. Is the manuscript technically sound, and do the data support the conclusions?

Reviewer #1: Yes

Reviewer #2: Yes

3. Has the statistical analysis been performed appropriately and rigorously? 

Reviewer #1: Yes

Reviewer #2: Yes

4. Have the authors made all data underlying the findings in their manuscript fully available?

Reviewer #1: (No Response)

Reviewer #2: Yes

5. Is the manuscript presented in an intelligible fashion and written in standard English?

Reviewer #1: Yes

Reviewer #2: Yes

6. Review Comments to the Author

Reviewer #1: The R1 addressed all the concernings I raised. Please pay attention about the scientific name while proofing.

Reviewer #2: Dear authors,

This study on the genetic basis of the cumulation of FAAs and TP was very significant. The experiment design was reasonable and the research method was advanced too, so the conclusion was reliable.

Thank you for your explaination for my comments. I have no other suggestion.

7. PLOS authors have the option to publish the peer review history of their article (what does this mean?). If published, this will include your full peer review and any attached files.

Reviewer #1: No

Reviewer #2: No

---

## [Editor Report · Acceptance letter]

22 Nov 2024

PONE-D-24-22074R1 

PLOS ONE

Dear Dr. Wang, 

I'm pleased to inform you that your manuscript has been deemed suitable for publication in PLOS ONE. Congratulations! Your manuscript is now being handed over to our production team.

Kind regards, 

on behalf of

Dr. Mojtaba Kordrostami 

Academic Editor

PLOS ONE